# Autophagy Contributes to Metabolic Reprogramming and Therapeutic Resistance in Pancreatic Tumors

**DOI:** 10.3390/cells11030426

**Published:** 2022-01-26

**Authors:** Gabriela Reyes-Castellanos, Nadine Abdel Hadi, Alice Carrier

**Affiliations:** Centre de Recherche en Cancérologie de Marseille (CRCM), CNRS, INSERM, Institut Paoli-Calmettes, Aix Marseille Université, F-13009 Marseille, France; grc989@hotmail.com (G.R.-C.); nadine.abdelhadi@inserm.fr (N.A.H.)

**Keywords:** pancreatic ductal adenocarcinoma, autophagy, cancer metabolism, mitochondrial metabolism, therapeutic resistance

## Abstract

Metabolic reprogramming is a feature of cancers for which recent research has been particularly active, providing numerous insights into the mechanisms involved. It occurs across the entire cancer process, from development to resistance to therapies. Established tumors exhibit dependencies for metabolic pathways, constituting vulnerabilities that can be targeted in the clinic. This knowledge is of particular importance for cancers that are refractory to any therapeutic approach, such as Pancreatic Ductal Adenocarcinoma (PDAC). One of the metabolic pathways dysregulated in PDAC is autophagy, a survival process that feeds the tumor with recycled intracellular components, through both cell-autonomous (in tumor cells) and nonautonomous (from the local and distant environment) mechanisms. Autophagy is elevated in established PDAC tumors, contributing to aberrant proliferation and growth even in a nutrient-poor context. Critical elements link autophagy to PDAC including genetic alterations, mitochondrial metabolism, the tumor microenvironment (TME), and the immune system. Moreover, high autophagic activity in PDAC is markedly related to resistance to current therapies. In this context, combining autophagy inhibition with standard chemotherapy, and/or drugs targeting other vulnerabilities such as metabolic pathways or the immune response, is an ongoing clinical strategy for which there is still much to do through translational and multidisciplinary research.

## 1. Introduction

Pancreatic cancer remains a poor-outcome disease with mortality rates nearly identical to incidence rates. It is expected to become the second cause of cancer mortality in the US in 2030, due to an increase in incidence and failure of treatments. Thus, there is an imperative need to develop effective therapeutic strategies against this incurable disease. This requires a better knowledge of its specific biology to identify new vulnerabilities amenable for targeting.

Therapeutic resistance stems mostly from remarkable metabolic plasticity of tumor cells conferring adaptation to the chemotherapy-induced stress and protection by the surrounding stroma. In that context, autophagy is a key metabolic process involved in stress resistance both intrinsically in tumor cells, and extrinsically in the tumor microenvironment (TME). In this review, we present the state of knowledge of autophagy as a process conferring high aggressiveness to pancreatic tumors, and the underlying cellular and molecular mechanisms.

## 2. Key Features of Pancreatic Ductal Adenocarcinoma (PDAC)

PDAC is the most frequent type of pancreatic cancer (90%). It remains a devastating disease with poor prognosis and limited efficiency of commonly available therapies [1,2]. PDAC is the fourth leading cause of cancer-related deaths in Western societies with a 5-year overall survival rate of only 10% [3,4]. Due to its late diagnosis, high metastatic capacity, aggressive local progression, and increase in incidence, PDAC is predicted to become the second cause of cancer deaths by 2030 in the US [4,5,6].

PDAC is a malignant epithelial tumor that arises from the exocrine portion of the pancreas and is mainly found in the head of the organ. Microscopically, PDAC consists of atypical tubular glands, partially formed or poorly formed glands with solid areas, depending on whether it is a well, moderate, or poorly differentiated neoplasm, respectively [7]. Importantly, PDAC develops from macroscopic and microscopic precursor lesions designated intraductal papillary mucinous neoplasms (IPMNs) and pancreatic intraepithelial neoplasias (PanINs) (Figure 1), respectively, the latter being believed to develop and progress asymptomatically over several decades [7,8]. Despite the ductal appearance of PanINs, it has been shown that acinar cells are at the origin of these lesions. Under specific conditions such as pancreatitis or genetic alterations, acinar cells can transform into duct-like structures, termed acinar–ductal metaplasia (ADM) [9]. ADM frequently progress to the well-characterized PanINs, that evolve to pancreatic tumors over time (Figure 1). Thus, ADM lesions can be considered as the main origin of PDAC [10].

The most frequent genetic alterations observed in PDAC are *KRAS*, *TP53*, *CDKN2A*, and *SMAD4* mutations, that promote tumorigenesis (Figure 1). Activating oncogenic mutations in *KRAS* are considered the major genetic initiating event in PDAC carcinogenesis and are present in 92–95% of cases [11,12]. They can be present in early premalignant lesions ADM and PanINs [9,13,14]. Subsequent mutations in the tumor suppressor genes *TP53*, *CDKN2A*, and *SMAD4*, occur later during disease progression and are also frequently observed with different percentages of cases (Figure 1). All these genetic alterations influence tumorigenesis affecting cell growth, cell metabolism, cell proliferation, and protein synthesis, collectively altering cellular homeostasis [15]. A main feature of PDAC is a high heterogeneity at tissular, genomic, epigenomic, and transcriptomic levels. Some histological subtypes of PDAC have been described as prognostic but not predictive of the effectiveness of anti-tumor treatments [16,17]. More recently, several molecular classifications have been proposed on the basis of genomic and transcriptomic data, and currently they are well accepted [11,16,18]. These classifications have so far contributed modestly to the improvement in therapeutic management [16,17]. PDAC is also characterized by an inflammation-induced desmoplastic reaction that results in a dense fibrotic stroma, consisting of abundant extracellular matrix (ECM) and stromal cells in the TME (Figure 1). This TME is very heterogeneous in terms of the variety of stromal cell subtypes, which includes cancer-associated fibroblasts (CAFs), immune cells, stroma-associated pancreatic stellate cells (PSCs), adipocytes, endothelial cells, and neurons. PDAC stroma cooperates with pancreatic tumor cells in a dynamic way to promote all aspects of PDAC aggressiveness including metastasis and intrinsic resistance to conventional therapies and immunotherapy [19,20,21,22].

PDAC is often diagnosed at late stage when it is a locally advanced–unresectable tumor or a metastatic disease. Consequently, for most PDAC patients, 80–85% of tumors are not resectable at the time of presentation and can only be treated with palliative chemotherapy [1,2]. In these patients, standard regimens include cytotoxic therapies, such as nab-paclitaxel plus gemcitabine, FOLFIRINOX (5-fluorouracil, leucovorin, irinotecan and oxaliplatin) or radiotherapy, and provide only modest increase in survival [23]. In contrast, in PDAC patients with resectable tumors (10–15%), adjuvant treatments with modified FOLFIRINOX (with a modification to exclude bolus doses of 5-fluorouracil) have contributed to the greatest improvements in survival [24,25]. However, a significant number of these patients will eventually develop a recurrent disease a few months after resection [26,27].

A great number of recent reports have indicated the important role of cancer metabolism in the therapeutic management of pancreatic cancer [28]. Interestingly, several studies also demonstrated that metabolic alterations can promote pancreatic tumorigenesis, metastasis, and chemoresistance, emphasizing the critical role of metabolism in pancreatic cancer development [8,29]. Metabolic reprogramming is a hallmark of cancer cells which regained considerable interest during the last two decades [30,31]. Cancer cells also depend on extensive metabolic interactions with other nonmalignant cells and elements from the TME [28]. Besides the rewiring of metabolic pathways within cells, cancer cells have multiple other mechanisms by which they acquire enough fuels for survival and growth. One of these mechanisms is autophagy, an essential cellular process allowing cells to recycle altered organelles and cellular components.

## 3. Autophagy and Tumor Metabolism

### 3.1. Autophagy: A Stress Response Process

Autophagy, meaning “self-eating”, is a protective response to different kinds of stress, in which cell metabolism uses the recycling of intracellular components to sustain cell survival [32,33,34,35,36,37]. Autophagy provides energy and metabolic substrates and supports stress response pathways. There are three forms of autophagy: macroautophagy, microautophagy, and chaperone-mediated autophagy (CMA). Macroautophagy (hereafter called autophagy) is the most studied form of autophagy and the topic of this review. Autophagy is regulated by more than 40 autophagy-related genes (Atg), which are well-conserved throughout the evolution from yeast to mammals [37,38].

A basal level of autophagy is found in most tissues to maintain protein and organelle quality. This occurs through the elimination of damaged intracellular material produced by the constitutive activity of cell functioning and metabolism (so-called “everyday stress”). Autophagy is highly upregulated in response to additional stress and starvation. Interestingly, multiple types of cancer cells can be considered as permanently stressed cells, showing in particular a high level of reactive oxygen species (ROS), which can explain why autophagy is high in certain cancer cells [39,40,41,42].

The autophagy process serves both quality control and recycling of cell components. It consists of the engulfment of cytosolic material—including organelles—into double-membrane vesicles, the autophagosomes, which then fuse with lysosomes delivering enzymes and an acidic environment for macromolecule (proteins, lipids, carbohydrates, nucleic acids) digestion in autolysosomes (Figure 2). The generated substrates are then used for the production of energy in mitochondria and synthesis of macromolecules to cope with the stress. Some selective types of autophagy target specific organelles, such as mitochondria (mitophagy), peroxisomes (pexophagy), the endoplasmic reticulum (reticulophagy), and lipid droplets (lipophagy), among others [38].

### 3.2. Autophagy: One of the Metabolic Pathways Reprogrammed in Cancer Cells

The dual role of autophagy in cancer has been the subject of intense research in recent years. It is now well demonstrated that autophagy is anti-tumoral in normal cells by preventing their transformation through elimination of damaged materials, whereas it promotes cancer cell survival in established tumors [34,36,43,44].

At the initial stages of tumor development, cancer cells located inside the tumor mass face nutrient depletion and hypoxia, triggering a stress response, including autophagy, that permits cell survival [45]. Only cells that adapt their metabolism to these harsh conditions can then proliferate and contribute to tumor growth. Thus, metabolic reprogramming, which appears at very early steps of tumorigenesis, is one of the features that are common in cancer cells, supporting cell survival and aberrant proliferation [31,45]. Along with the rewiring of several metabolic pathways, autophagy is dysregulated in cancer cells, some of them developing a dependence for autophagy (so-called “autophagy-addicted cancer cells”) [46,47]. Some mechanisms of this dependence are dysregulated signaling pathways stemming from alterations of oncogenes such as RAS and tumor suppressors like p53 [46,48,49].

The Warburg effect or aerobic glycolysis, which is the first reported cancer metabolic alteration, occurs early during tumor growth, being induced by the Hypoxia-Inducible Factor (HIF) [45]. Contrary to the previously accepted dogma of cancer cells’ reliance on glycolysis for energy (ATP) production, mitochondria maintain their activity in most cancer cells. They are the major source of ATP by the oxidative phosphorylation (OXPHOS) in the respiratory chain, along with their main roles as metabolic hubs and in redox and apoptosis control [29,50,51]. Mitochondrial respiration can occur at oxygen concentrations as low as 0.5%, thus it can still function even in hypoxic environments during tumor development and inside tumors [31]. Mitochondria are sensitive to the stress conditions associated with initial tumor growth, including the oxidative stress induced by hypoxia; indeed, ROS can damage mitochondrial membranes by oxidation. In turn, stressed mitochondria with altered OXPHOS produce higher levels of ROS through increased partial reduction of oxygen at the level of respiratory complexes I and III. As a response, autophagy is induced by ROS, and damaged mitochondria over-producing ROS are eliminated by mitophagy [42,52]. In parallel, autophagy is able to promote mitochondrial metabolism [53,54]. Thus, autophagy and mitochondrial metabolism collaborate in cancer cells to sustain their survival and proliferation in favor of tumor growth [55,56].

### 3.3. Autophagy Is Highly Active in Various Types of Cancer Cells

In recent years, autophagy was demonstrated to be upregulated in many different types of cancer, including pancreatic cancer (see below). Autophagy is required to support metabolism, tumorigenesis, and survival in harsh conditions within tumors and during therapy [37,43,46,57,58,59]. Interestingly, high autophagic activity is not restricted to cancer cells in tumors. Autophagy in stromal cells is one of the alternative ways by which cancer cells can acquire nutrients from their TME [46,58,60,61]. Other alternative ways enable tumor cells to scavenge nutrients from the TME cells and extracellular matrix, which are abundant in several tumors such as PDAC. These alternative ways involve uptake of extracellular proteins via macropinocytosis, entosis of living cells, and phagocytosis of dead cells and apoptotic corpses [59,62]. The crosstalk between autophagy and other alternative ways of acquiring nutrients is presented in Figure 2. Moreover, autophagy in tumor-infiltrating immune cells is affected by multiple stress signals, thereby modifying their survival, activation, and fate in the TME [61].

Some cancer cells are addicted to autophagy, meaning that they rely on autophagy for survival [46,47]. In addition, most of the chemotherapeutic drugs induce cellular stress, thus increasing autophagic activity, which participates in drug resistance [37]. These observations drive the idea of inhibiting autophagy in combination with chemotherapy as a therapeutic strategy in the clinic [57,61]. Clinical studies using Chloroquine (CQ) or Hydroxychloroquine (HCQ), which inhibit the last step of autophagy, showed disappointing results, supporting the interest of developing more specific drugs targeting earlier steps of autophagy [57,61].

## 4. Autophagy Is a Key Feature of PDAC Metabolism

As mentioned previously, pancreatic cancer TME presents a dense desmoplastic stroma, which limits oxygen, nutrients, and drug delivery to the cells. However, how can pancreatic cancer cells survive and proliferate in such adverse conditions? In fact, pancreatic cancer cells are well adapted to these conditions by several mechanisms, including the use of unorthodox strategies for nutrient acquisition [8,63]. Among these, scavenging and recycling pathways are by far the most frequently cited, highlighting the role of autophagy in sustaining PDAC. In most cancers, autophagy is low under basal conditions and is activated as a survival response upon chemotherapy or other stress insults. However, it has been 10 years now since Kimmelman’s lab demonstrated that PDAC tumors exhibit constitutive autophagy, making this mechanism an essential requirement for PDAC growth and maintenance [64]. Therefore, pancreatic tumors are highly sensitive to autophagy inhibitors in vitro and in vivo, resulting in ongoing clinical trials and several drug-development strategies [64,65].

Notwithstanding, the role of autophagy in pancreatic cancer seems to be more complex and context dependent, with reports indicating both pro- and anti-tumorigenic effects (Figure 1). In PDAC, autophagy can prevent PDAC initiation at early steps of the disease as briefly explained in the next section. On the other hand, in established tumors, autophagy supports tumor growing and maintenance by different mechanisms. Given this complex panorama, it is necessary to further study why pancreatic tumors show this peculiarity that leads to an “autophagy addiction” in established tumors. This review mainly focuses on the pro-tumoral role of autophagy in pancreatic cancer.

### 4.1. Autophagy Has an Anti-Tumoral Role at Early Steps of PDAC Tumorigenesis

Even if autophagy is mainly considered as a survival pathway to resist stress conditions in cancer, paradoxically, autophagy suppression during early stages of PDAC may be pro-tumoral. Basal autophagy maintains the proper acinar cell function to sustain the high protein synthetic rates of these cells. Loss of autophagy induces acinar cell dysfunction leading to pancreatitis and spontaneous activation of regenerative mechanisms that initiate ADM, as demonstrated using a mouse model lacking the essential autophagy gene *Atg7* [66]. Interestingly, constitutive autophagy is also present in the pancreatic β-cells to maintain homeostasis, and autophagy dysfunction causes insulin deficiency and hyperglycemia [67]. Therefore, dysregulated autophagy may play a role in the pathogenesis of diabetes, one of the well-known risk factors of pancreatic cancer.

Furthermore, Rosenfeldt and collaborators demonstrated that in mice with oncogenic *Kras*, the lack of essential autophagy genes induced low-grade PanIN lesions, but not PDAC development, confirming the anti-tumoral role of autophagy at early steps of tumorigenesis [68]. However, in mice with concomitant oncogenic *Kras* and loss of p53, autophagy suppression induced PanIN transformation into invasive PDAC. This work provided substantial insight into this topic and in a therapy context, since the authors showed that pharmacological inhibition of autophagy accelerated tumor formation in mice containing oncogenic *Kras* but lacking p53, instead of providing a benefit. However, contrary to these findings, Yang and collaborators revealed that the response to autophagy inhibition is independent from the p53 status [69,70].

Besides its well-known role in tumor suppressive processes such as cell cycle arrest, senescence, and apoptosis, p53 is also involved in autophagy, metabolism, and redox control [48,71]. Our laboratory identified the p53 target Tumor Protein 53-Induced Nuclear Protein 1 (TP53INP1) as a tumor suppressor and a new actor in autophagy [72,73]. We demonstrated that TP53INP1 plays a crucial role in the cellular antioxidant defense through both regulating p53 activity in the nucleus and participating in the elimination of altered mitochondria by mitophagy in the cytoplasm [74,75,76,77]. We also showed that TP53INP1 is lost in early stages of PDAC development before the acquisition of p53 mutations, and that this loss is associated with oxidative stress promoting oncogenic KRAS-driven tumorigenesis [78,79,80]. Collectively, these observations suggest that the loss of TP53INP1 favors PDAC development through the lack of autophagy which protects against oxidative stress.

Finally, autophagy functions in removing misfolded proteins, damaged organelles, and protein aggregates, and therefore provides the cell with an important quality control mechanism. This control activity serves as a barrier to tumorigenesis through suppression of genomic instability, oxidative stress, and chronic tissue damage. However, it is important to notice that established cancers can use this protective function of autophagy to gain a growth advantage and protect the tumor cells [8].

### 4.2. Autophagy Supports PDAC Progression at Late Steps of Tumorigenesis and Growth of Established Tumors

Pancreatic cancer metabolism is widely reprogrammed as in other cancers [29], and the high dependence on autophagy is a clear example of this metabolic adaptation required for continued growth of established pancreatic tumors. In this line, Guo and collaborators showed that RAS activation—a key genetic event in PDAC—renders cancer cells autophagy-dependent, especially in starvation conditions [81]. However, it should be noted that KRAS activation is an early event in PDAC and maintained during cancer progression, thus ruling out that KRAS is the only determinant involved with the autophagy reliance observed in PDAC.

Moreover, other works revealed the complexity of autophagy in PDAC, showing that autophagy is upregulated even from high grade PanINs and in metastatic tumors [64], which comprises tumor cell-intrinsic and -extrinsic mechanisms [65]. Below some of the most important elements that link autophagy to pancreatic cancer are described and illustrated (Figure 3), these elements being a result of the continuous harsh conditions to which tumor cells are subjected.

#### 4.2.1. Autophagy Sustains Mitochondrial Metabolism to Meet Biosynthetic and Bioenergetics Demands

Evidence from White’s group showed that autophagy is required to preserve mitochondrial function, and specifically, autophagy supports the growing of cancer cells with active RAS mutations, including the pancreatic cancer cell line Panc-1 [81]. In this study, the knockdown of essential autophagy regulators (Atg5 and Atg7) in RAS-activated cancer cells impaired mitochondrial function, reflected by loss of mitochondrial membrane potential and reduced levels of mitochondrial respiration, cellular energy, and tricarboxylic acid (TCA) cycle metabolites. Importantly, the levels of TCA metabolites produced only by mitochondria (citrate, aconitate, and isocitrate) significantly dropped in cells with defective autophagy under both basal and starvation conditions, which was not the case for TCA metabolites that can be generated also in the cytosol. The authors concluded that RAS activation renders cancer cells dependent on autophagy, which in turn supports cancer survival via mitochondrial function. Interestingly, we demonstrated that primary pancreatic cancer cells display high OXPHOS heterogeneity, which was unrelated to KRAS mutations [82]. Moreover, deletion of an enzyme essential for autophagy (Atg7) in several cell lines with oncogenic KRAS mutations showed no inhibition of cell proliferation and tumorigenesis in vivo [83]. This suggests that KRAS mutations do not predict cell-autonomous addiction to autophagy.

Products generated by autophagy are used to fuel biosynthetic and bioenergetic reactions, these fundamental reactions being largely carried out by mitochondria. This statement is supported by the work of Guo and colleagues in KRAS-driven lung tumors. They demonstrated that autophagy-deficient cancer cells are sensitive to starvation due to impaired mitochondrial substrate supply. Moreover, the authors revealed that autophagy induced by starvation supplies specific substrates, glutamine and glutamate being critical to feed the TCA cycle to preserve mitochondrial function [84]. Accordingly, exogenous supplementation of glutamine and glutamate enables survival of autophagy-deficient cancer cells during starvation. Finally, this work showed that starved cancer cells with autophagy loss presented a profuse reduction in the total nucleotide pools, which was restored after nucleoside supplementation.

In lung tumors with concomitant KRAS activation and p53 loss, autophagy dysfunction (by Atg7 deletion) converts tumors to oncocytomas. These benign tumors are characterized by defective mitochondrial respiration and lipid accumulation, reflected in a reduced Fatty Acid Oxidation (FAO) and increased sensitivity to FAO inhibition [85]. In this line, autophagy regulates FAO to sustain OXPHOS in acute myeloid leukemia (AML) via mitochondria–endoplasmic reticulum (ER) contact sites (MERCs) [86]. In another study, Kamphorst and collaborators showed that lysophospholipids scavenging is a major route of fatty acid acquisition in both hypoxia and RAS-driven cancer cells, however, the mechanism by which cells scavenge these lipids remained to be elucidated [87]. These studies are instrumental to understand the relation between autophagy and the different metabolic inputs that power mitochondria in cancer. Recently, our laboratory demonstrated that mitochondrial respiration of pancreatic tumors depends mainly on the fatty acids to meet basal energetic demands (Reyes-Castellanos et al., unpublished), and autophagy may be a source of fatty acids for mitochondria.

#### 4.2.2. Interplay between Pancreatic Cancer Cells and Host Autophagy

Autophagy is not exclusive to tumoral cells, and there is compelling evidence that this process occurring in cells from the TME or from distant organs (“host autophagy”) can influence tumor growth [37]. In this context, the work of Sousa and collaborators showed that the predominant cell type in PDAC TME, the PSCs, carry out autophagy to provide alanine as an alternative carbon source to support mitochondrial metabolism and tumor growth [60]. Notably, alanine is a direct mitochondrial carbon source as observed by the increased mitochondrial respiration (and not glycolysis), feeding the TCA cycle for non-essential amino acids (NEAA) and lipid biosynthesis. Interestingly, pancreatic cancer cells can stimulate the PSCs for autophagy induction and the release of alanine. In addition, increased autophagy causes the PSCs to change from a dormant state to an active state [88]. Activated PSCs secrete ECM molecules and IL-6 to increase the aggressiveness of pancreatic cancer.

Autophagy in distant tissues is implicated in cancers [37]. In the recent study of Yang and collaborators, systemic inhibition of autophagy impacted tumor engraftment, supporting the contribution of non-cell autonomous effects in the efficacy of autophagy inhibition [65]. Importantly, autophagy in skeletal muscle is a key proteolysis pathway which is activated during PDAC-induced cachexia and sarcopenia (loss of muscle mass and strength), suggesting that it may participate in the feeding of PDAC tumors with amino-acids originating from the sarcopenic muscle [89,90,91,92].

#### 4.2.3. Autophagy Protects from Immune Elimination

The pro-tumoral function of autophagy is attributed in great part to its cytoprotective capacity in tumor cells subjected to common stresses during cancer progression [93]. The classical view of autophagy implies that it serves as a mechanism for cancer cells to cope with different stressors; however, pancreatic cancer cells exhibit high levels of autophagy even under basal conditions, suggesting that they are chronic-stressed cells. This may indicate also that autophagy could have additional biological significance in PDAC. In this line, autophagy could be a means to evade immune elimination resulting in reduced efficacy of immunotherapy [63]. Accordingly, Yamamoto and collaborators demonstrated that in pancreatic cancer cells, selective autophagy targets the major histocompatibility complex class I (MHC-I) for degradation in lysosomes [94]. Hence, PDAC cells present with decreased MHC-I expression at their cell surface, affecting antigen presentation. In another study, a functional–genomic screening revealed that autophagy enabled pancreatic cancer cells to evade CD8^+^ T cell killing via Tumor Necrosis Factor-alpha (TNFα)-induced cell death [95]. Interestingly, Poillet-Perez and colleagues demonstrated that host autophagy, specifically in hepatocytes, suppresses anti-tumor T-cell response and IFN-γ production promoting the growing of high tumor mutational burden (TMB) tumors [96]. Thus, autophagy helps cancer cells, including PDAC cells, to evade the anti-tumoral immune response.

#### 4.2.4. The Dual Role of Autophagy in Invasion and Metastasis

Metastasis development requires that cells survive several biological challenges: break away from the primary tumor, survival in the circulation, colonization of distant organs, and growth into tumors at these remote sites [30]. Consequently, metastasis formation imposes metabolic requirements that cancer cells must meet to survive and grow, and here autophagy becomes critical. However, it is important to clarify that in some cases autophagy can act as a suppressor of metastasis thus having a dual role [97]. For instance, in a study analyzing various levels of Atg5 in PDAC, complete loss of Atg5 blocks tumorigenesis, while monoallelic loss of Atg5 promotes tumorigenesis and metastasis development. The authors suggest some elements to be crucial for PDAC aggressiveness, including changes in mitochondrial homeostasis. Accordingly, they found that with loss of one Atg5 allele, mitochondrial function was reduced, mitochondrial stress markers were upregulated, and an increase in fission events and mitophagy was observed. Interestingly, in human PDAC samples, lower Atg5 expression levels are associated with tumor migration and shorter patient survival. This suggests that the Atg5 expression levels should be considered when using autophagy inhibitors to treat PDAC, to avoid the occurrence of drug resistance and highly aggressive cells [98].

The epithelial–mesenchymal transition (EMT) is a process in which epithelial cells acquire interstitial characteristics that participate in promoting metastatic events [97]. In the PDAC Panc-1 human cell line and other cancer cells, ECM detachment induces autophagy, which in turn protects from detachment-induced cell death (anoikis) and facilitates glycolysis, promoting adhesion-independent transformation driven by oncogenic KRAS [99]. However, in some cases autophagy inhibition may specifically activate the EMT program in KRAS-mutated PDAC cells by triggering the NF-κB pathway via SQSTM1/p62 [100].

### 4.3. Autophagy Contributes to Therapeutic Resistance

A characteristic of pancreatic cancer is its highly therapeutic resistance due to some key features that are detailed in the review of Beatty and collaborators [63]. Among these features, metabolic aberrations, including constitutive autophagy, contribute to the therapeutic resistance of PDAC. Twenty years ago, Paglin and collaborators were pioneers in demonstrating that autophagy is a defense mechanism against radiotherapy in different cancers [101]. Since then, several studies have confirmed this statement in different therapies and cancers [37]. Nowadays, it is clear that autophagy is activated to target damaged organelles for destruction and to regulate signaling programs, as, for example, the damage induced to mitochondria or DNA by chemotherapy and radiotherapy [63]. In the next paragraphs we present important mechanisms allowing therapeutic resistance via autophagy in pancreatic cancer.

#### 4.3.1. Autophagy as a Compensatory Mechanism following KRAS Pathway Inhibition

Pharmacological inhibition of the RAF→MEK→ERK signaling pathway (downstream of KRAS) using Trametinib activates autophagy, protecting PDAC cells from the cytotoxic effects of this inhibition [102]. Mechanistically, the authors suggest that Trametinib-induced autophagic flux serves to protect PDAC cells from the potentially pro-apoptotic effects (cell death evasion) of the pathway inhibition. This observation is consistent with the work reported by Bryant and collaborators, in which pharmacological inhibition of ERK1/ERK2 increases autophagic flux in *KRAS*-mutant PDAC cell lines [103]. ERK inhibition stimulated AMPK activation, and decreased mTORC1 signaling, which are well-known mechanisms that elicit autophagy. Moreover, ERK inhibition increases autophagy at the levels of autophagic signaling, nucleotide metabolism, and gene transcription. Notably, they demonstrated that either *KRAS* suppression or ERK inhibition decreased both glycolytic and mitochondrial functions, and autophagy may function as a compensatory mechanism. Collectively, this work demonstrates that ERK inhibition elicits autophagy via multiple mechanisms. To summarize, these findings can be instrumental to understand, at least in part, why targeting the KRAS pathway has provided no clinical benefit to patients to date.

#### 4.3.2. Cell Death Evasion Linked to Mitophagy

The ability to resist cell death is a hallmark of cancer cells and can be regulated by autophagy that depends on stimulation [97]. In the case of PDAC, autophagy is a major mechanism used by cancer cells to evade apoptosis following treatment with chemotherapy [104], contributing to chemotherapeutic resistance. Concretely, an altered expression of the apoptosis-regulating molecules—including members of the Bcl-2 family—is a common feature [105]. In this line, mitophagy is a specific form of autophagy that selectively degrades damaged mitochondria and is highly regulated by the pro-apoptotic Bcl2/adenovirus E1B 19 kDa protein-interacting protein 3 (BNIP3). Mitophagy is mainly induced by mitochondrial depolarization, hypoxia, and metabolic stress, including that caused by chemotherapeutics [52]. Due to the antitumoral role of BNIP3 linked to mitophagy, downregulation of this pro-apoptotic protein is associated with chemoresistance of pancreatic cancer cell lines to gemcitabine and 5-fluoro-uracil [105,106].

#### 4.3.3. Cancer Stem Cells (CSCs) and Autophagy

Several studies have considered the CSCs as the main subpopulation responsible for chemoresistance, including in pancreatic cancer [107]. A predominant mechanism of chemoresistance induced by CSCs is the overexpression of ATP-binding cassette (ABC) transporters, and this correlates with autophagy [108]. In an in vitro study, forcing pancreatic cancer cells to use OXPHOS resulted in CSCs enrichment, then these cells overexpressed the ABC subfamily G member 2 (ABCG2) leading to reduced intracellular concentrations of drugs and increased survival under treatment with different chemotherapeutics, including gemcitabine [109]. Notably, these CSCs cells showed increased autophagy, suggesting that autophagy may be an important mechanism used by OXPHOS-dependent cells. Intriguingly, these CSCs cells also exhibited immune evasion, a property provided by autophagy as described before.

Collectively, the mechanism by which autophagy promotes therapeutic resistance is likely to be multifaceted. However, evidence indicates that metabolism, mainly oxidative, is implicated in most of these mechanisms.

## 5. Therapeutic Approaches Using Autophagy Inhibitors in PDAC

Autophagy has been shown to be elevated in PDAC and it is implicated in resistance to both cytotoxic chemotherapy and targeted therapy [65]. Within this framework, the essential role of autophagy in promoting pancreatic cancer has made it a promising therapeutic target in PDAC [110,111]. Clinical interventions to manipulate autophagy in cancer therapy are already under investigation, with the vast majority focused on autophagy inhibition [46]. The most widely employed compounds that inhibit the last stage of autophagy are Chloroquine (CQ) and its derivative Hydroxychloroquine (HCQ), Bafilomycin A1 (BafA1), and lysosomal protease inhibitor cocktails. CQ and HCQ are synthetic 4-aminoquinolines initially developed for malaria disease and thus are the only autophagy inhibitors approved by the Food and Drug Administration (FDA) [112,113]. Therefore, their effects on several cancers, including pancreatic, have been deeply studied as explained below.

### 5.1. Preclinical Trials

Since autophagy is a complex and context-dependent process in PDAC as in other cancers, its targeting has to be well supported by preclinical data regarding the role and status of autophagy in a particular cancer [114]. In this regard, several preclinical studies support the idea that autophagy inhibition may improve PDAC treatment in a clinical setting. A large number of in vitro assays and in vivo studies using genetically-engineered mouse models (GEMMs) and patient-derived xenograft (PDX) have been conducted [46]. Albeit genetic interventions in autophagy are widely used in cancer research, pharmacological inhibition is more kinetically controllable, and is the most frequently employed approach in preclinical studies [112].

Due to their elevated autophagy, PDAC cells are highly sensitive to autophagy inhibition. Accordingly, CQ markedly decreased the proliferation and anchorage-independent growth in a panel of PDAC cell lines but had minimal impact on other cancer cells with low basal autophagy flux [64]. Moreover, autophagy inhibition with CQ has a potent impact in vivo since it can both eradicate PDAC growth and prolong survival in subcutaneous and orthotopic xenografts and in a KRAS-driven GEMM [64]. Importantly, in the subcutaneous model, tumors seemed to have a sustained complete regression in half of the CQ-treated mice [64]. Interestingly, in a large panel of PDX with *TP53* mutations, HCQ treatment attenuated the growth of most xenografts, showing the clinical relevance of autophagy inhibition independent of p53 status [70].

The impact of autophagy inhibition with CQ has also been tested in pancreatic CSCs, resulting in decreased CSCs populations, sphere formation, and resistance to gemcitabine in vitro and in vivo. Consequently, autophagy blockade sensitized pancreatic CSCs to gemcitabine and enhanced its efficacy against PDAC. Clearly, in this work a high coexpression of LC3 and ALDH1 correlated with poor prognosis, and this was mediated by osteopontin (OPN)/NF-κB signaling [115]. Although HCQ has shown improvement in PDAC studies, there has been controversy about its off-target effect, particularly the inhibition of other aspects of lysosomal scavenging. Therefore, Yan and colleagues developed an inducible mouse model (Atg4B mutant) that allowed the acute and reversible inhibition of autophagy. Excitingly, this inhibition caused significant tumor regression, being a result of both tumor cell-intrinsic and host effects [65].

Given that KRAS mutations are almost ubiquitous to PDAC, the National Cancer Institute has identified the development of anti-KRAS therapies as one priority for pancreatic cancer research. One approach is to target KRAS-dependent metabolic functions, including autophagy. Paradoxically, ERK inhibition in PDAC impaired glycolysis and mitochondrial metabolism, leading to increased autophagy. However, ERK1/ERK2 inhibition in association with CQ enhanced the autophagy-inhibitory activity of the latter in vitro, blocking tumor progression and extending survival using HCQ in KRAS-mutant pancreatic subject-PDX [103]. In a similar way, inhibition of MEK1/2 with Trametinib in combination with CQ resulted in a synergistic anti-proliferative effect against PDAC cell lines and induced regression of xenografted pancreatic tumors [102].

Immunotherapy has not been successful in pancreatic cancer, being largely refractory to immune checkpoint inhibitors (ICI) [116]. However, systemic autophagy inhibition with CQ, as well as tumor-specific autophagy inhibition, sensitizes PDAC to immune blockade. This occurs via restoring MHC-I surface levels and enhancing anti-tumor CD8^+^ T cell responses in immunocompetent host mice [94]. Therefore, targeting autophagy may promote the efficacy of immunotherapy and overcome immunotherapy resistance [61].

It is important to mention that CQ has been reported to present anti-tumor effects independent of autophagy, as shown by Eng et al., where Atg7-deficient and -proficient cells were equally sensitive to the antiproliferative effect of CQ [83]. This demands appropriate genetic stratification parameters to predict efficacy towards this compound. Finally, besides autophagy inhibitors, it will be important to determine the efficacy of autophagy inducers in combination with chemotherapeutics, immunotherapy, or MAPK inhibitors [63].

### 5.2. Clinical Trials

The development of clinical trials is a result of a vast number of preclinical studies, some of them with promising results, such as those works showing complete tumor regression of PDAC tumors in different mouse models. Due to its clinical availability and well-recognized activity as an autophagy inhibitor, HCQ has been tested in several clinical trials in association with chemotherapy or radiotherapy [52].

In pancreatic cancer, most clinical trials targeting autophagy used HCQ in combination with standard chemotherapies or MAPK inhibitors. In these trials, besides the objective of improving patient survival, other clinical endpoints include the serum carbohydrate antigen (CA) 19-9 biomarker [113,117], histopathologic response [113], and immune response [113,118], important for assessing high risk tumors, margins after surgery, and immunomodulatory effects, respectively. Targeting autophagy could potentially enhance immunotherapy, and clinical trials using HCQ in combination with immunotherapy are ongoing to treat patients with different types of cancer [61]. In the case of pancreatic cancer, a sole clinical trial is published using a combinatorial strategy of HCQ/gemcitabine/nab-paclitaxel/avelumab (NCT03344172), however, it was terminated due to suspected serious adverse effects related to the treatment.

Importantly, it is recognized that CQ and HCQ can have several off-target effects, thus Zeh and colleagues analyzed autophagic markers in resected PDAC tumors, finding the sequestosome protein SQSTM1/p62 accumulation associated with HCQ treatment [113]. Interestingly, in this study, no difference was found for the common recycled autophagy marker LC3B II/Atg8 between controls and treated tumors. Still, in different types of solid cancers, the ability of HCQ to inhibit autophagy, together with its safety, has been demonstrated supporting its use for treating cancer [119].

Table 1 recapitulates the completed or ongoing clinical trials using CQ or HCQ as autophagy inhibitors in the treatment of pancreatic cancer.

## 6. Conclusions

This review recapitulates strong evidence that the constitutive activation of autophagy in PDAC tumors is a major mechanism of aggressiveness and therapy resistance in patients. Autophagic dependence should be monitored at diagnosis for each tumor, to unveil autophagy activity as a vulnerability to be targeted. This knowledge would serve to better define the best therapeutic approach, combining autophagy inhibition with standard chemotherapy, and/or drugs targeting other vulnerabilities such as metabolic pathways or the immune response [28,29,80,122].

## Figures and Tables

**Figure 1 cells-11-00426-f001:**
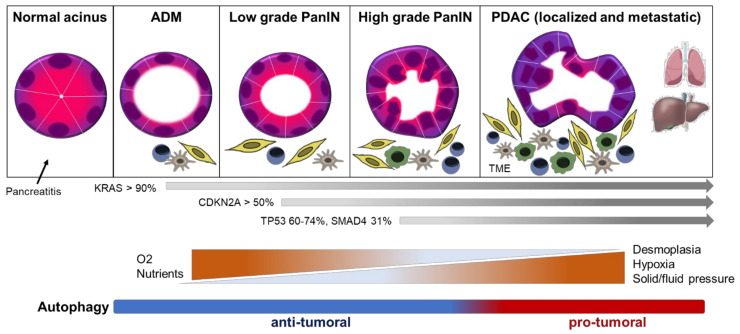
Schematic representation of Pancreatic Ductal Adenocarcinoma (PDAC) carcinogenesis, the main genetic events involved, and key metabolic features. PDAC is a malignant epithelial neoplasm that arises from the exocrine portion of the pancreas, mainly from the acinar cells. Under severe stress conditions such as pancreatitis, acini can transform into duct-like structures, termed acinar–ductal metaplasia (ADM). ADM frequently progress to the well-characterized pancreatic intraepithelial neoplasias (PanINs), defined as mucinous-papillary proliferations with a ductal appearance. These lesions can be classified into low or high grade PanINs based on the degree of architectural and nuclear atypia. Genomic mutations of pancreatic cancer predominate in four genes, the *KRAS* oncogene and the tumor suppressor genes *CDKN2A*, *TP53*, and *SMAD4*. KRAS oncogenic mutations are nearly ubiquitous in PDAC and occur early in carcinogenesis, and more importantly, these point mutations drive constitutive KRAS activation thus maintaining cell proliferation and survival. Subsequent mutations in the tumor suppressor genes are present later further contributing to disease progression. Besides genetic alterations, PDAC is characterized by a prominent fibrotic stroma (desmoplasia), consisting of abundant extracellular matrix (ECM) and stromal cells in the tumor microenvironment (TME). This TME is very heterogeneous in terms of the variety of stromal cell subtypes, which includes cancer-associated fibroblasts (CAFs, in yellow), immune cells such as lymphocytes (blue) and macrophages (green), stroma-associated pancreatic stellate cells (PSCs, in gray), among others. This prominent TME exerts high levels of solid and fluid pressure, and compression of vasculature, limiting oxygen and nutrient availability during PDAC progression. Notwithstanding, pancreatic cancer cells are well adapted to these adverse conditions by several mechanisms, including autophagy, which shows both pro- and anti-tumorigenic effects depending on the context. In PDAC, autophagy can prevent cancer initiation at early steps of the disease, and in established tumors, autophagy supports PDAC growing and maintenance by different mechanisms.

**Figure 2 cells-11-00426-f002:**
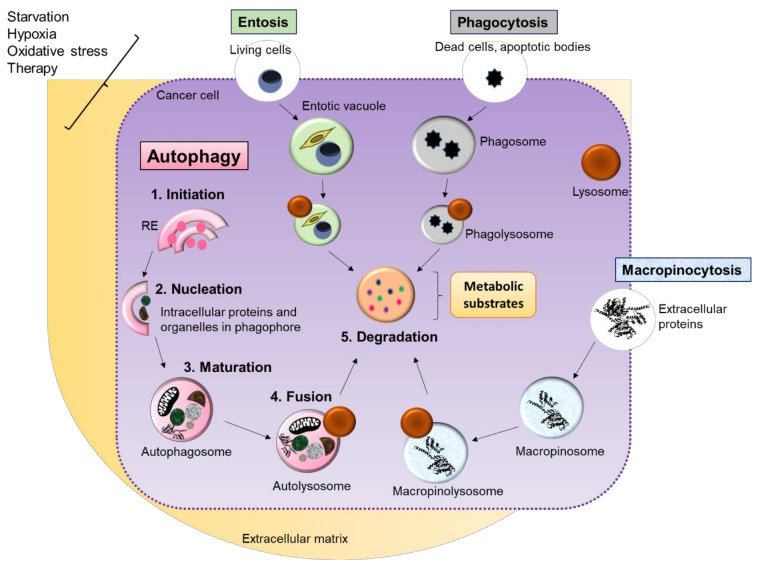
Autophagy and other scavenging ways of nutrient acquisition in cancer cells. Autophagy consists of the recycling of intracellular material (proteins and damaged organelles) to sustain cell survival. This process includes five steps: (1) initiation; (2) nucleation; (3) maturation; (4) fusion; (5) degradation. At first, different stressors such as starvation and hypoxia induce the breaking of the endoplasmic reticulum (ER) to form the phagophores. After, material is engulfed into double-membrane vesicles, the autophagosomes, which then fuse with lysosomes (autolysosomes) to deliver enzymes and acidification for macromolecule degradation. Other ways that enable tumor cells to scavenge nutrients from the TME and extracellular matrix include the uptake of proteins via macropinocytosis, entosis of living cells, and phagocytosis of dead cells and apoptotic bodies. These scavenging ways present similar key steps to the autophagy process, in which a mature structure containing material fuses with lysosomes to finally degrade macromolecules for generation of metabolic substrates.

**Figure 3 cells-11-00426-f003:**
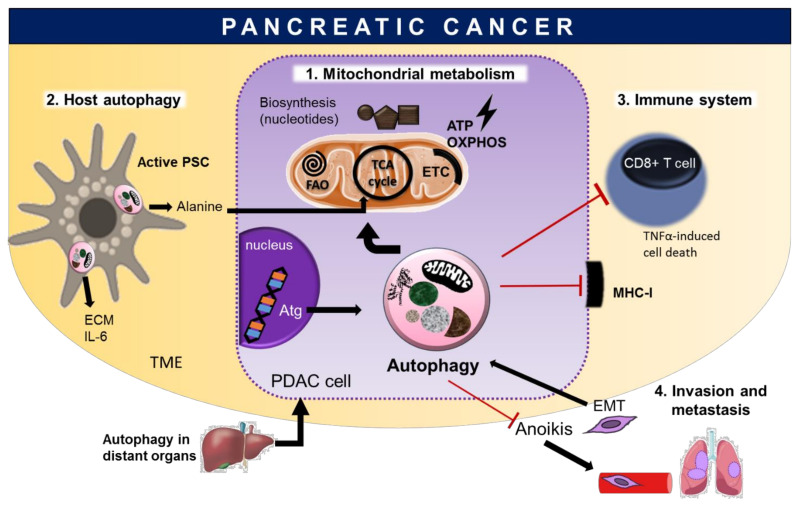
Autophagy supports Pancreatic Ductal Adenocarcinoma (PDAC) progression at late steps of tumorigenesis. PDAC cells exhibit high autophagy even under basal conditions and some elements are key to understand how autophagy sustains established pancreatic tumors. **1. Mitochondrial metabolism****.** Autophagy is necessary for a proper mitochondrial function, in particular for feeding the tricarboxylic acid (TCA) cycle with recycled substrates and for synthesis of nucleotides and macromolecules (biosynthesis). Moreover, autophagy can allow substrates such as fatty acids to enter the Fatty Acid Oxidation (FAO) that feeds the TCA cycle to power the Electron Transport Chain (ETC) for energy production (bioenergetics), via Oxidative Phosphorylation (OXPHOS). **2. Host autophagy****.** Autophagy occurring in both cells from the Tumor Microenvironment (TME) or distant organs can influence tumor growing. PDAC cells activate the stroma-associated pancreatic stellate cells (PSCs) for inducing autophagy to provide alanine as an alternative carbon source to support mitochondrial metabolism and tumor growth. In addition, activated PSCs secrete extracellular matrix (ECM) molecules and interleukin-6 (IL-6) to increase the aggressiveness of pancreatic cancer. Moreover, systemic inhibition of autophagy impacts tumor growing, supporting the contribution of non-cell autonomous effects in the efficacy of autophagy inhibition. **3. Immune system****.** Autophagy assists PDAC immune evasion. In PDAC, selective autophagy targets the major histocompatibility complex class I (MHC-I) for degradation in lysosomes. Hence, PDAC cells present with decreased MHC-I expression at their cell surface, affecting antigen presentation. Furthermore, autophagy enables pancreatic cancer cells to evade from CD8^+^ T cell killing via Tumor Necrosis Factor-alpha (TNFα)-induced cell death. **4. Invasion and metastasis development****.** Epithelial–Mesenchymal Transition (EMT) is a characteristic of malignancy. EMT detachment induces autophagy, which in turn protects from detachment-induced cell death (anoikis) and facilitates glycolysis, promoting adhesion-independent transformation.

**Table 1 cells-11-00426-t001:** Clinical trials targeting autophagy in pancreatic cancer.

Autophagy Inhibitor	Additional Treatment	PDAC Stage	Clinical Response (Primary Endpoint)	Study Phase	Recruitment Status	ClinicalTrials.Gov Identifier
HCQ	n/a	Metastatic (previously treated)	PFS at 2 months: 10%	II	Completed	NCT01273805, ref. [118]
	**In combination with cytotoxic chemotherapies**					
CQ	Gemcitabine	Unresectable or metastatic	Median OS: 7.6 months	I	Completed	NCT01777477, ref. [120]
HCQ	Gemcitabine plus nab-Paclitaxel	Potentially resectable tumors	Histopathologic response: Improved with HCQ (*p* = 0.00016)	II	Completed	NCT01978184, ref. [113]
HCQ	Gemcitabine plus nab-Paclitaxel	Advanced or metastatic	OS at 1 year: 41% (HCQ) vs. 49% (controls)	I/II	Active, not recruiting	NCT01506973, ref. [117]
HCQ	Gemcitabine	Resectable (preoperative)	OS (months): 34.83 vs. 12.27 (controls)	I/II	Completed	NCT01128296, ref. [121]
HCQ	Paricalcitol with Gemcitabine plus nab-Paclitaxel	Advanced or metastatic	n/a	II	Recruiting	NCT04524702
HCQ	Capecitabine plus radiation	Resectable	n/a	II	Active, not recruiting	NCT01494155
HCQ	Gemcitabine/ nab-Paclitaxel/Avelumab	Resectable	Suspected adverse events related to treatment	II	Terminated	NCT03344172
	**In combination with MAPK pathway inhibitors**					
HCQ	Trametinib (MEK inhibitor)	Various	n/a	I	Recruiting	NCT03825289
HCQ	Binimetinib (MEK inhibitor)	Metastatic	n/a	I	Recruiting	NCT04132505
HCQ	LY3214996 (ERK inhibitor)	Metastatic	n/a	II	Recruiting	NCT04386057
HCQ	Ulixertinib (ERK inhibitor)	Advanced	n/a	I	Recruiting	NCT04145297

CQ, chloroquine; HCQ, hydroxychloroquine; OS; overall survival; PFS, progression-free survival.

## Data Availability

Not applicable.

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
