# Peer review of "Autophagy Contributes to Metabolic Reprogramming and Therapeutic Resistance in Pancreatic Tumors"

_cells, 2022, doi:10.3390/cells11030426_

Round 1

Reviewer 1 Report

This manuscript reviews recently published papers dealing with autophagy in PDAC and discusses potential therapeutic strategies for PDAC from the aspect of cellular metabolism as an important mechanism promoting tumor development. It wouldn’t be easy to summarize recent advances in the autophagy in PDAC, but I think this manuscript will serve as a good introduction for anyone working on autophagy and/or pancreatic cancer.

However, I found some statements that may mislead the reader, which are listed below. Also, some sentences and paragraphs are somewhat difficult to understand and sound redundant, so English editing is strongly recommended before accepting. As for Figure 1, I think it is redundant in its present form and could mislead the reader. First, it is not clear how autophagy is involved in each phenomenon. Second, autophagy occurs inside cells, but the figure ignores this location. Thus, it should be revised or deleted.

Line 123: “autophagolysosomes” should be “autolysosomes”

Line 127: “lipophagy” is a type of selective autophagy for lipid droplets, but not “metabolites such as lipids”. So, the sentence should be collected.

Line 154: “ , (or) with ROS damaging…”?

Line 166: “TME” has been explained in the earlier part.

Line 167-169: How are “alternative ways “related with autophagy?

Line 172: “increase the rate of autophagy”, what does this mean?

Line 174-175: The sentence “In addition, …” should be accompanied by some references.

Line 177-179: The sentence “Clinical assays …” should be accompanied by some references.

Line 177: “clinical assay” should be “clinical trials” or “clinical studies”?

Line 196-197: The sentence “In PDAC …” should be accompanied by some references

Line 209: Is “ADM” an abbreviation? It appears only here.

Line 210: “…the essential autophagy-related protein 7 (Atg7)” should be “…the essential gene, Atg7”, since the abbreviation “Atg” has been mentioned in the text. I think copying the sentence in the reference may cause a misleading.

Line 234: “… through lack of protective antioxidant autophagy” needs to be reworded.

Line 294: I think the phrase “In our laboratory” is redundant.   

Line 301: “… these fundamental reactions being carried out by mitochondria” I think this sentence cause a misunderstanding, because mitochondria is not the only organelle for those events.

Line 304: “starving autophagy-deficient cells” should be rephrased.

Line 304: “Accordingly… “ I think some explanations are required before mentioning this sentence.Overall, I do not well understand this paragraph.

Line 306: “p53 loss deficiency”?

Line 369: increased?

Line 399-408: This part summarizes one paper, while two literatures are cited. Although one of them is a review article, readers would be puzzled.

Line 421-423: I do not understand this sentence. “Contrary to…”, why?

Line 455: “may improve PDCA”?

Line 430: Some explanations are required for the term, “ABCG2”.

Line 461-466: This part requires some references.

Line 464: “… with low basal (autophagy?) flux”

Author Response

Point-by-point Response to Reviewer 1

This manuscript reviews recently published papers dealing with autophagy in PDAC and discusses potential therapeutic strategies for PDAC from the aspect of cellular metabolism as an important mechanism promoting tumor development. It wouldn’t be easy to summarize recent advances in the autophagy in PDAC, but I think this manuscript will serve as a good introduction for anyone working on autophagy and/or pancreatic cancer.

However, I found some statements that may mislead the reader, which are listed below. Also, some sentences and paragraphs are somewhat difficult to understand and sound redundant, so English editing is strongly recommended before accepting. As for Figure 1, I think it is redundant in its present form and could mislead the reader. First, it is not clear how autophagy is involved in each phenomenon. Second, autophagy occurs inside cells, but the figure ignores this location. Thus, it should be revised or deleted.

Response: We warmly thank the Reviewer for appreciating our review and pointing out that it will be helpful for anyone working on autophagy and/or PDAC. We apologize for the misleading statements and the difficulty faced during the reading of some paragraphs. We tried our best to improve the English of the text and to better explain the paragraphs where needed. Concerning Figure 1, we totally understand the issue raised by the reviewer, so we tried to improve the figure and to illustrate that autophagy occurs inside the cells. However, we think that to add more information about each phenomenon will considerably saturate the figure causing confusion to the reader.

In addition, it is important to mention that we added two new figures, one showing the PDAC progression, key genetic events and metabolic features, including autophagy. Moreover, Figure 2 takes advantage of the question raised by reviewer 2 regarding the relation between autophagy and other ways of nutrient acquisition. We think that these three figures are now worthful to illustrate the text, and that the two additional Figures (Figures 1 and 2) allow a better understanding of previous Figure 1 (now Figure 3).

Line 123: “autophagolysosomes” should be “autolysosomes”

Response: We did the change (in line 166 now).

Line 127: “lipophagy” is a type of selective autophagy for lipid droplets, but not “metabolites such as lipids”. So, the sentence should be collected.

Response: We did the correction in the text (in line 170 now).

Line 154: “ , (or) with ROS damaging…”?

Response: We did improve the clarity of the sentence (in line 198 now).

Line 166: “TME” has been explained in the earlier part.

Response: We did define TME for the first time in the introduction section (line 40). In the rest of the text, the abbreviation is used.

Line 167-169: How are “alternative ways “related with autophagy?

Response: We thank the reviewer for this interesting question. Indeed, we prepared a Figure 2 illustrating the autophagy process by itself and its relation with other ways of nutrient acquisition (macropinocytosis, entosis, and phagocytosis) (lines 215-218 now). We hope that the Reviewer will appreciate this addition.

Line 172: “increase the rate of autophagy”, what does this mean?

Response: We did modify the sentence for: “In addition, most of chemotherapeutic drugs induce cellular stress, thus increasing autophagic activity, which participates in drug resistance” (lines 238-239 now).

Line 174-175: The sentence “In addition, …” should be accompanied by some references.

Response: We added a reference (#37, Poillet-Perez L. et al. Autophagy is a major metabolic regulator involved in cancer therapy resistance. Cell Rep 2021, 36, 109528) (line 240 now).

Line 177-179: The sentence “Clinical assays …” should be accompanied by some references.

Response: We did modify the text (summarizing two sentences in one), the resulting sentence is now accompanied with two references (line 245 now).

Line 177: “clinical assay” should be “clinical trials” or “clinical studies”?

Response: We thank the reviewer for this correction. We changed for “clinical studies” (line 242 now).

Line 196-197: The sentence “In PDAC …” should be accompanied by some references

Response: We thank the reviewer for this suggestion. However, since these statements are explained in the sections below with references, we decided to not add them at this point. We did precise this in the sentence (in line 263 now).

Line 209: Is “ADM” an abbreviation? It appears only here.

Response: We thank the reviewer to point out this. Since acinar-ductal metaplasia (ADM) is key to understand PDAC tumorigenesis, we defined this concept in section 2 (line 60 now). In the rest of the text, such as here (line 274 now), we use only the abbreviation, except in the legend of Figure 1, where we consider that it is important to define ADM.

Line 210: “…the essential autophagy-related protein 7 (Atg7)” should be “…the essential gene, Atg7”, since the abbreviation “Atg” has been mentioned in the text. I think copying the sentence in the reference may cause a misleading.

Response: We did the change as suggested (line 275 now).

Line 234: “… through lack of protective antioxidant autophagy” needs to be reworded.

Response: The sentence now is reworded and clearer (line 301 now).

Line 294: I think the phrase “In our laboratory” is redundant.

Response: We deleted “in our laboratory” (line 364 now).

Line 301: “… these fundamental reactions being carried out by mitochondria” I think this sentence cause a misunderstanding, because mitochondria is not the only organelle for those events.

Response: We totally agree with this comment and thus we added the word “largely” to be more precise. The sentence now is: “these fundamental reactions being largely carried out by mitochondria” (line 371 now).

Line 304: “starving autophagy-deficient cells” should be rephrased.

Response: We did modify by “autophagy-deficient cancer cells during starvation” (line 379 now).

Line 304: “Accordingly… “ I think some explanations are required before mentioning this sentence.Overall, I do not well understand this paragraph.

Response: We agree with the reviewer and we apologize for the lack of better explanation. We did better develop the findings of this article in a larger and clearer paragraph (lines 372-381 now).

Line 306: “p53 loss deficiency”?

Response: We did eliminate “deficiency” (line 383 now).

Line 369: increased?

Response: We did correct by “increase” (line 449 now).

Line 399-408: This part summarizes one paper, while two literatures are cited. Although one of them is a review article, readers would be puzzled.

Response: We agree with the reviewer comment; thus, we suppressed the review and only kept the original article (reference #103) (line 482 now).

Line 421-423: I do not understand this sentence. “Contrary to…”, why?

Response: We did correct this mistake. “Contrary” was replaced with “due” (line 502 now).

Line 455: “may improve PDCA”?

Response: Sorry for the mistake. The sentence is now “may improve PDAC treatment” (line 538 now).

Line 430: Some explanations are required for the term, “ABCG2”.

Response: We did better define ABC transporters and ABCG2 (lines 509-512 now).

Line 461-466: This part requires some references.

Response: We thank the reviewer for the suggestion, however since this paragraph explains mainly a sole article, we decided to keep the only reference (but citing it in each related sentence), except in the last sentence, where there is the explanation of another reference (lines 544-553 now).

Line 464: “… with low basal (autophagy?) flux”

Response: Certainly, we added “autophagy” (line 547 now).

Reviewer 2 Report

This review summarizes succinctly the state of the field as it applies to the role that autophagy plays in reprogramming metabolism in pancreatic cancer and its contribution to therapy resistance. The manuscript is clearly written and covers important topics. 

There are some minor editing that need to be addressed that mainly have to do with English editing and clarity. These are summarized below:

Line 36: consider replacing the word “stands” with the word “stems” and the word “important” with the word “remarkable”, so the sentence reads “Therapeutic resistance stems mostly from the remarkable metabolic plasticity of tumor cells …”

Line 124: replace the word “for” with the word “with” so it reads “… to cope with the stress.”

Line 187: replace “frequent” with “frequently” (cited, …)

Line 197: Replace the word “In” with “On” (the other hand…)

Line 204: the word “contrarrest” should be replaced with “resist”

Lines 228-235: have a different font type than the rest of the text

Line 247: consider replacing “it is to remember” with “it should be noted”

Line 248: delete “all the” preceding “cancer progression”

Line 354: delete the word “from” following the word “evade” so the sentence reads “… cancer cells evade CD8+ T cell killing via …”

Sentence 355-356: “Thus, autophagy is a metabolic requirement for PDAC immune evasion” – this sentence is not supported by the summary in the preceding paragraph. There is no connection made between autophagy-regulated metabolism and immune evasion. The paragraph simply makes the point that elevated autophagy help PDAC cells evade immune cells by downregulating MHC-1 or TNF.

Line 362: replace the word “growth” with “grow”

Line 364: replace “By” with “For”, so the sentence begins with “For instance …”

Line 527-528: consider replacing part of the sentence “… it has been demonstrated the ability of HCQ to inhibit autophagy, together with its safety, supporting its use for treating cancer” with ““…the ability of HCQ to inhibit autophagy, together with its safety, has been demonstrated supporting its use for treating pancreatic cancer”.

Line 538: replace “would serve better defining” with “would serve to better define” (the best therapeutic approach …)

Author Response

Point-by-point response to Reviewer 2

This review summarizes succinctly the state of the field as it applies to the role that autophagy plays in reprogramming metabolism in pancreatic cancer and its contribution to therapy resistance. The manuscript is clearly written and covers important topics.

Response: We warmly thank the reviewer 2 for appreciating our manuscript and for the following minor suggestions.

There are some minor editing that need to be addressed that mainly have to do with English editing and clarity. These are summarized below:

Line 36: consider replacing the word “stands” with the word “stems” and the word “important” with the word “remarkable”, so the sentence reads “Therapeutic resistance stems mostly from the remarkable metabolic plasticity of tumor cells …”

Response: We really appreciate the suggestions of the reviewer to improve the clarity of the paper. We did the suggested changes.

Line 124: replace the word “for” with the word “with” so it reads “… to cope with the stress.”

Response: We did the replacement (line 168 now).

Line 187: replace “frequent” with “frequently” (cited, …)

Response: We did the change (line 252 now).

Line 197: Replace the word “In” with “On” (the other hand…)

Response: We made this replacement (line 263 now).

Line 204: the word “contrarrest” should be replaced with “resist”

Response: We made this replacement (line 269 now).

Lines 228-235: have a different font type than the rest of the text

Response: We did homogenize the font type. Thank you very much.

Line 247: consider replacing “it is to remember” with “it should be noted”

Response: We made this replacement (line 313 now).

Line 248: delete “all the” preceding “cancer progression”

Response: We did delete it (line 315 now).

Line 354: delete the word “from” following the word “evade” so the sentence reads “… cancer cells evade CD8+ T cell killing via …”

Response: We did the deletion (line 430 now).

Sentence 355-356: “Thus, autophagy is a metabolic requirement for PDAC immune evasion” – this sentence is not supported by the summary in the preceding paragraph. There is no connection made between autophagy-regulated metabolism and immune evasion. The paragraph simply makes the point that elevated autophagy help PDAC cells evade immune cells by downregulating MHC-1 or TNF.

Response: We thank to the reviewer for this important point, which we corrected. The sentence now is: “ autophagy helps cancer cells, including PDAC cells, to evade the antitumoral immune response” (lines 434-435 now). We did this change also in the legend of Figure 3 (previous Figure 1).

Line 362: replace the word “growth” with “grow”

Response: We made this replacement (line 442 now).

Line 364: replace “By” with “For”, so the sentence begins with “For instance …”

Response: We did the replacement (line 444 now).

Line 527-528: consider replacing part of the sentence “… it has been demonstrated the ability of HCQ to inhibit autophagy, together with its safety, supporting its use for treating cancer” with ““…the ability of HCQ to inhibit autophagy, together with its safety, has been demonstrated supporting its use for treating pancreatic cancer”.

Response: We really appreciate the suggestions of the reviewer to improve the clarity of the paper. We did the changes (lines 610-612 now).

Line 538: replace “would serve better defining” with “would serve to better define” (the best therapeutic approach …)

Response: We did the replacement (line 622 now).